# Semi-Supervised Semantic Segmentation-Based Remote Sensing Identification Method for Winter Wheat Planting Area Extraction

**Mingmei Zhang** [1], **Yongan Xue** [2], **Yuanyuan Zhan** [3] **and Jinling Zhao** [3,*]

1 Department of Geological and Surveying Engineering, Shanxi Institute of Energy, Jinzhong 030600, China; zhangmm@sxie.edu.cn
2 College of Mining Engineering, Taiyuan University of Technology, Taiyuan 030024, China; xueyongan@tyut.edu.cn
3 National Engineering Research Center for Agro-Ecological Big Data Analysis & Application, Anhui University, Hefei 230601, China; p20201094@stu.ahu.edu.cn
* Correspondence: zhaojl@ahu.edu.cn

**Abstract:** To address the cost issue associated with pixel-level image annotation in fully supervised semantic segmentation, a method based on semi-supervised semantic segmentation is proposed for extracting winter wheat planting areas. This approach utilizes self-training with pseudo-labels to learn from a small set of images with pixel-level annotations and a large set of unlabeled images, thereby achieving the extraction. In the constructed initial dataset, a random sampling strategy is employed to select 1/16, 1/8, 1/4, and 1/2 proportions of labeled data. Furthermore, in conjunction with the concept of consistency regularization, strong data augmentation techniques are applied to the unlabeled images, surpassing classical methods such as cropping and rotation to construct a semi-supervised model. This effectively alleviates overfitting caused by noisy labels. By comparing the prediction results of different proportions of labeled data using SegNet, DeepLabv3+, and U-Net, it is determined that the U-Net network model yields the best extraction performance. Moreover, the evaluation metrics MPA and MIoU demonstrate varying degrees of improvement for semi-supervised semantic segmentation compared to fully supervised semantic segmentation. Notably, the U-Net model trained with 1/16 labeled data outperforms the models trained with 1/8, 1/4, and 1/2 labeled data, achieving MPA and MIoU scores of 81.63%, 73.31%, 82.50%, and 76.01%, respectively. This method provides valuable insights for extracting winter wheat planting areas in scenarios with limited labeled data.

**Keywords:** semi-supervised classification; sematic segmentation; winter wheat; self-training; data augmentation

## 1. Introduction

Semantic segmentation is a fundamental task in the field of computer vision, and it has made significant progress in many application areas [1–3]. Fully supervised semantic segmentation learns to assign pixel-level semantic labels by generalizing from a large number of densely annotated images. Despite rapid progress, the cost of manually annotating pixels is much higher compared to other visual tasks such as image classification and object detection. Reliable pixel-wise segmentation annotations are typically only available for a few classes and images, making fully supervised semantic segmentation challenging for tasks involving diverse objects. To further simplify the process of acquiring high-quality data, semi-supervised semantic segmentation has been proposed, which learns models from a small number of labeled images and a large number of unlabeled images to achieve the final segmentation goal. Many researchers have applied semi-supervised semantic segmentation to crop extraction. Casado-García et al. [4] used a pseudo-label

semi-supervised learning method combined with a semantic segmentation network to segment natural color images of vineyards captured by cameras for plant yield monitoring. Zheng et al. [5] improved segmentation accuracy with a small number of labeled data by using a semi-supervised adversarial semantic segmentation network for building extraction on three different resolution datasets (WBD, MBD, and GID). Mukhtar et al. [6] proposed a semi-supervised method based on cross-consistency for semantic segmentation of RGB images captured by drones and further counted the extracted plant clusters, providing insights for studying other crops (such as rice and maize) using a small number of labeled images. In the field of semi-supervised semantic segmentation, two widely used forms of semi-supervised methods are entropy minimization and consistency regularization [7].

Self-training [8,9], also known as self-supervised learning, is a semi-supervised learning technique where a model iteratively generates pseudo-labels for unlabeled data and then uses these pseudo-labeled samples to retrain itself. It aims to improve the model's accuracy by iteratively refining its own predictions, often by considering high-confidence predictions as pseudo-labels. Zhu et al. [10] achieved optimal performance with less supervision using the self-training semi-supervised method on the Cityscape, CamVid, and KITTI datasets. Feng et al. [11] proposed a dynamic self-training and class-balanced curriculum algorithm for semi-supervised semantic segmentation. They achieved robustness to pseudo-label noise in self-training by weighting the pixel-wise loss based on predicted confidences and providing a pseudo-label curriculum to gradually label all unlabeled data, demonstrating good performance on PASCAL VOC 2012 and Cityscapes. Zoph et al. [12] compared self-training with ImageNet pre-training and revealed the universality and flexibility of self-training. These studies serve as typical examples of self-training methods. However, this approach also has certain limitations. If the teacher model trained based on labeled data contains errors, it can lead to erroneous results in the subsequent student model.

Recently, methods based on consistency regularization have been proposed, which rely on the fact that unlabeled and labeled data share the same distribution. Therefore, these methods aim to train models that have consistent and reliable predictions for both labeled and unlabeled data. One widely researched approach is to use different unlabeled images under different transformations as inputs to the model and enforce consistency loss on the predicted masks. In consistency regularization methods [13,14], consistency is enhanced by augmenting input images, feature representations, and networks to ensure consistency with various perturbations, such as input perturbations, which involve perturbing unlabeled data [15,16]. Based on the clustering assumption, data points with different class labels are separated from each other, while similar data points have similar outputs. Therefore, if an unlabeled data point is perturbed and consistency constraints are enforced on the predicted masks of the perturbed images, the predicted results should not change significantly. Common transformations include data augmentation techniques such as adding noise, cropping, and scaling [17], as well as methods like CutMix [18] and ClassMix [19]. These form the basis of consistency-based methods and many self-supervised learning methods, all of which focus on leveraging unlabeled data. For example, the Π-model parameterizes input samples with different noises and adds a regularization term to reduce the difference between the outputs of the perturbed samples and the original inputs. Temporal ensembling [20] and mean teacher [21] involve ensemble learning, using exponentially moving average (EMA) weights to improve the quality of labels for perturbed samples. CCT (cross-consistency training) [22] is a semi-supervised semantic segmentation method based on cross-consistency, which applies perturbations to the input of the encoder instead of directly perturbing the input, exploring the application of perturbations at different network layers in the segmentation network. Interpolation consistency training (ICT) achieves a modification in the perturbation method by using a mixture with another unlabeled sample instead of adding random noise, which is considered more effective in dealing with low-margin unlabeled points. Berthelot et al. [23] further propose sharpening artificial labels for unlabeled data and using MixUp to mix labeled and unlabeled data.

Generative adversarial networks (GANs) are generally utilized for tasks related to generative modeling, where they generate data that are similar to the training data. They indirectly might benefit semi-supervised learning but not as a direct supervision signal, are not easy to optimize, and may encounter mode collapse issues [24]. Pseudo-labeling [25] is a combination of entropy minimization and consistency regularization, which relies on the assumption that pseudo-labels generated by a teacher model can benefit the training of a new model. FixMatch [26] achieves consistency training by applying strong data augmentation (Cutout, CTAugment, and RandAugment) to the unlabeled loss and input for pseudo-label generation, encouraging the outputs of two augmented inputs to remain consistent. As an extension of FixMatch, PseudoSeg [27] uses weak and strong augmentations on an unlabeled image and uses the predictions of the weakly augmented image as pseudo-labels for the strongly augmented image. This method generates pseudo-labels using only the image itself.

In conclusion, semi-supervised learning offers a significant reduction in the annotation process, resulting in resource savings in terms of manpower and materials. Nevertheless, it can be found that the performance of these semi-supervised semantic methods can be limited by the quality and quantity of the labeled data available. Inaccurate or inconsistent annotations can hinder the learning process and result in suboptimal segmentation performance. In addition, some methods may require complex architectures or training procedures, which can make them computationally expensive and difficult to implement in real-world applications. It is necessary to reduce the reliance of segmentation methods on high-quality pseudo-labels while simultaneously minimizing the number of iterative rounds required to generate such labels. Currently, most related studies have focused on publicly available datasets, with limited research on remote sensing crop extraction. This paper aims to extract winter wheat planting areas using a semi-supervised semantic segmentation method that combines entropy minimization and consistency regularization. The proposed approach utilizes a deep learning semantic segmentation network to achieve the desired results. By eliminating the need to filter high-quality pseudo-labels and generating them only once for unannotated images, significant reductions in training time can be achieved.

## 2. Materials and Methods

### 2.1. Study Area

The selected study area (Figure 1) is Zhengding County and Zengcun Town, Gaocheng District, Shijiazhuang City, Hebei Province, China, located at approximately 37°51′~38°18′ N, 114°39′~114°59′ E. The study area has a temperate semi-humid to semi-arid continental monsoon climate, with distinct seasons in most regions. The annual average temperature is 12.9 °C, and the annual average precipitation is 550 mm. Cultivated land is the predominant land use type in this region, with winter wheat being one of the major food crops.

### 2.2. Data Preprocessing and Label Generation

Based on the phenological characteristics of winter wheat, during the grain filling stage, its growth is more advanced compared to other crops, which are either not yet sown or have just been sown. At this stage, there is a significant difference between winter wheat and other land cover types, enabling the high-precision segmentation and extraction of wheat planting areas. Therefore, in this study, remote sensing imagery data were acquired during the milk-ripe stage of winter wheat, around mid-to-late May. A Landsat-8 OLI (operational land imager) image was acquired, and data preprocessing was also performed. After finishing the geometric and radiometric corrections, the image fusion with 15 m panchromatic and 30 m multispectral bands was carried out using the high-pass filtering fusion.

Subsequently, the labeled data were generated using the preprocessed image. Firstly, the remote sensing imagery was opened using ArcGIS Desktop 10.8 software. The wheat area was delineated on the original image, and a vector was generated to create labels in PNG format. In the labels, wheat areas were represented by white color with a pixel

value of 1, while non-wheat areas (background) were represented by black color with a pixel value of 0. Simultaneously, the original remote sensing imagery was saved as JPG format output. Both the original imagery data and label data were randomly cropped to 256 × 256 pixels. The file names of the original imagery data corresponded to the label data. Data augmentation operations were performed, including rotation, mirror operations, brightness adjustment, and adding noises (Gaussian noise, salt and pepper noise).

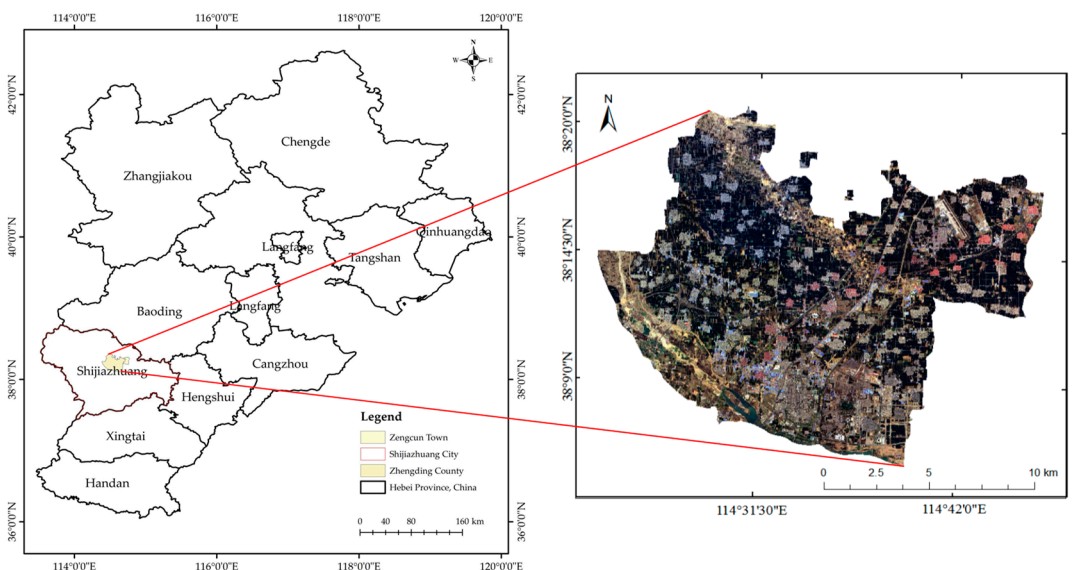

**Figure 1.** Geographical location of the study area.

### 2.3. Data Augmentation

Self-training, relying on an initial model trained with labeled data, does not attempt to address the negative impact of noisy pseudo-labels and is not well-suited for the sparse labeling mechanism in semi-supervised learning. Specifically, when iteratively overfitting to incorrect supervision, errors in pseudo-labels accumulate and significantly degrade the performance of the student model. Additionally, in this self-learning process, the introduced information during training is insufficient, resulting in a severe coupling problem between the teacher model and the retrained student model. As the predicted pseudo-labels in the second stage may still contain a considerable amount of noise, directly retraining on these images and labels containing noise can easily lead to overfitting to the noisy labels. Specifically, the student model is forced to learn the pseudo-labels from the teacher model in a supervised manner during retraining. However, due to the same network structure and similar initialization of the teacher and student models, they tend to make similar true and false predictions on unlabeled images, making it difficult for the student model to learn additional information beyond minimizing entropy during training.

To address the above issues, namely overfitting to noisy labels and the prediction coupling between the student and teacher models, data augmentation is applied to the unlabeled images during the retraining stage to propose more challenging optimization objectives for the student model. Data augmentation is an effective regularization technique, with basic strategies such as random flipping and cropping commonly used in training visual models. Classic augmentation perturbations, such as cropping, scaling, and rotation, were also used on the original images in the data set construction, and these basic perturbations as weak data augmentation are difficult to confuse output categories. Therefore, stronger and more diverse augmentations were used on the unlabeled images in semi-supervised semantic segmentation. Color transformation data augmentation was used, including Grayscale, Colorjitter, Blur, and RandomInvert, as well as spatial transformation using Cutout [28] (Figure 2).

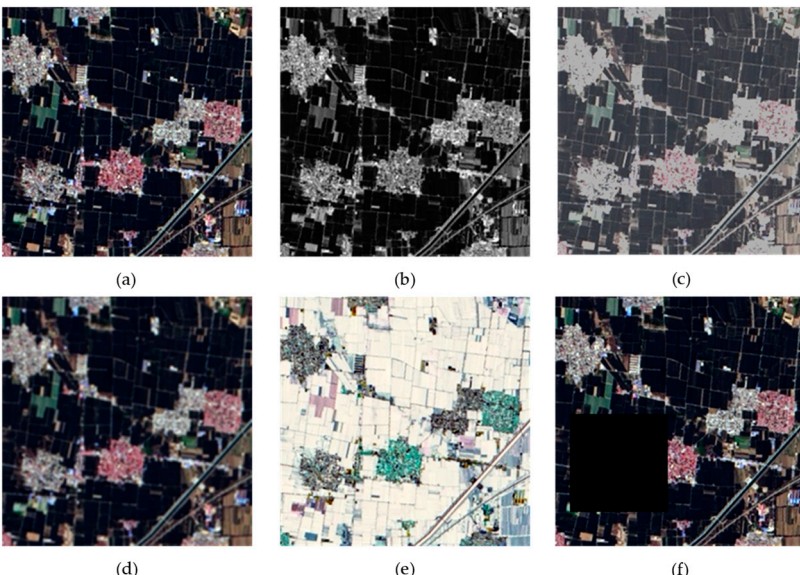

**Figure 2.** Comparison of different data augmentation methods: (**a**) Original image; (**b**) Grayscale; (**c**) Colorjitter; (**d**) Blur; (**e**) RandomInvert; and (**f**) Cutout.

### 2.4. Experimental Environment and Parameter Setting

The experimental setup consisted of an Intel Xeon Gold 6248R processor, 192 GB of memory, an NVIDIA Quadro P4000 graphics card, and the GPU acceleration library CUDA 10.0 with the PyTorch deep learning framework. For model training, the SGD optimizer was chosen as the parameter optimizer, with a base learning rate of 0.001, 80 training iterations, and a step size of 8. Weak data augmentation techniques, such as rotation, mirroring, and noise addition, were applied as discussed in the previous section. Strong data augmentation techniques on unlabeled images were implemented with the following settings: Grayscale random probability $p$ was set to 0.5; Colorjitter had brightness, contrast, and saturation values of 0.5 and a hue value of 0.25; Blur had a sigma value range of (0.1, 2.0); RandomInvert had a random probability $p$ of 0.5; and Cutout had a size value range of (0.02, 0.4).

### 2.5. Methodology

#### 2.5.1. Entropy Minimization

Entropy minimization [29] is an effective semi-supervised approach for achieving clustering assumptions, as it encourages more confident predictions for unlabeled data by forcing the classifier to make low-entropy predictions. Under the assumption that the majority of data points are far from the decision boundary, this method prevents the decision boundary of the network from being close to the data points; otherwise, it would be forced to produce less reliable predictions. This approach can be achieved by adding a loss term for predictions, as follows:

$$L = -\sum_{k=1}^{c} f_\theta(x)_k \log f_\theta(x)_k \tag{1}$$

where $k$ is the number of classes and $f_\theta(x)_k$ is the model's confidence in predicting whether $x$ belongs to class $k$. The sum of confidence for all classes is 1, meaning that when the prediction value for one class is close to 1, the prediction values for other classes are close to 0, resulting in minimized entropy.

Self-training is a form of entropy minimization in semi-supervised learning. Its main idea is to collect both labeled and unlabeled data, but only use the labeled data to train an initial teacher model. This model is then used to construct pseudo-labels for the unlabeled data, which are combined with the labeled data for joint training to obtain a new student

prediction model. However, self-training also has certain limitations. For example, the pseudo-labels generated based on the teacher model can be erroneous and unreliable, leading to a deterioration in the performance of the student model during iterative training. Therefore, in recent years, researchers have been exploring ways to improve the feasibility of self-training in order to enhance the accuracy and reliability of the model. One approach is to set a probability threshold, where pseudo-labels are only generated for unlabeled data when the prediction probability exceeds the threshold. Another approach is to use iterative training, where, after generating pseudo-labels for unlabeled data, a new supervised model is trained by combining the pseudo-labels and labeled data. This new model is then used for further predictions, allowing for iterative updates to correct previous erroneous pseudo-labels. Alternatively, training multiple teacher models can be employed, where only pseudo-labels recognized by the majority of teacher models are successfully generated.

2.5.2. Consistency Regularization

In the realm of semi-supervised semantic segmentation algorithms, the concept of consistency regularization has been widely employed to enhance segmentation accuracy [30]. By perturbing unlabeled data, the current optimization model is compelled to produce stable and consistent predictions for the same unlabeled data under different perturbations, such as variations in shape and color. Leveraging the assumption of clustering, the addition of perturbations does not significantly affect the actual output results. This approach does not require specific label information, making it suitable for semi-supervised learning. By constructing an unsupervised regularization loss term between the predictions obtained from perturbed and unperturbed versions of unlabeled data, the model's generalization ability is improved. Mathematically, the formulation is as follows:

$$\mathrm{D}[p_{\mathrm{model}}(y|\mathrm{Augment}(x), \theta), p_{\mathrm{model}}(y|\mathrm{Augment}(x), \theta] \tag{2}$$

where D represents a metric function, typically utilizing KL (Kullback–Leibler) divergence or JS (Jensen–Shannon) divergence, and can employ cross-entropy or mean squared error; $\mathrm{Augment}(\cdot)$ denotes a data augmentation function that introduces noise perturbations, and $\theta$ represents the model parameters.

To mitigate the potential errors and unreliability of pseudo-labels generated by the teacher model, methods to improve the quality of generated labels can be employed. There are two approaches to enhance label quality: rational selection of the teacher model and appropriate choice of input perturbations. In the study [21], the mean teacher architecture was utilized, which is a consistency regularization method based on the assumption of smoothness. By perturbing unlabeled data using CutMix, the overfitting problem of neural networks is reduced. This allows the model to consistently predict both the given unlabeled data and its perturbed versions. By combining labeled data and pseudo-labels, the student model's parameters are updated through backpropagation. The student model's parameters are exponentially moving and averaged to serve as the teacher model's parameters, continuously iterating to shape an improved teacher model. Consequently, the quality of the student model is continuously enhanced, resulting in improved segmentation accuracy.

CCT is a semi-supervised semantic segmentation method based on cross-consistency that achieves prediction consistency through various forms of perturbations applied to the encoder outputs [22]. By training a shared encoder and a main decoder with labeled data, multiple auxiliary decoders are trained to leverage unlabeled data. The different perturbed versions of the shared encoder's output are used as inputs, and consistency is enforced between the predictions of the main decoder and the auxiliary decoders. The encoder's representation is strengthened by training signals extracted from unlabeled data. Perturbations such as Cutout are applied to unlabeled images, resulting in higher segmentation accuracy on the PASCAL VOC 2012 dataset.

PseudoSeg continues the exploration of consistency regularization by applying different perturbations to images [27]. PseudoSeg employs two types of data augmentation on input images: weak augmentation (random cropping and flipping, for instance) and strong

augmentation (color jittering). Both augmented images are fed into the same network, producing two distinct outputs. Due to the stability of training under weak augmentation, confidence vectors are constructed from the outputs generated by weakly augmented images to generate pseudo-labels. Finally, the loss is computed using the pseudo-labels and the vectors obtained from strongly augmented images.

2.5.3. Self-Training Algorithm

Semi-supervised semantic segmentation [31] aims to generalize from a combination of pixel-labeled images of $D^l = \{(x_i, y_i)\}_{i=1}^M$ and unlabeled images of $D^u = \{u_i\}_{i=1}^N$, where, in most cases, $N \gg M$, and the overall optimization objective is formalized as follows:

$$L = L^s + \lambda L^u \tag{3}$$

Here, $\lambda$ is the weight between labeled and unlabeled data. The unsupervised loss is represented as $L^u$, while the supervised loss $L^s$ is the cross-entropy loss between the predicted and manually annotated masks.

It involves three steps and does not require iterative training (Algorithm 1):

(1) Supervised learning: Train a teacher model $T$ using the cross-entropy loss on a labeled dataset of $D^l$.
(2) Pseudo-labeling: Use the trained teacher model $T$ to predict one-hot pseudo-labels on an unlabeled dataset of $D^u$, resulting in $\hat{D}^u = \{(u_i, T(u_i))\}_{i=1}^N$.
(3) Retraining: Combine the labeled and pseudo-labeled data of $D^l \cup \hat{D}^u$ and retrain a student model $S$ for final testing.

Here, the unsupervised loss $L^u$ can be expressed as follows:

$$L^u = \mathrm{H}(T(x), S(A^w(x))) \tag{4}$$

Here, $x$ represents the input image, and $T$ and $S$ map the image $x$ to the output space. $A^w$ applies random, weak data augmentation to the original image. H minimizes the entropy between the student and teacher.

Due to the random, strong data augmentation applied to each unlabeled image before feeding it into the network model, it is highly beneficial for decoupling predictions of the same input and mitigating overfitting to noise pseudo-labels. In other words, although the images are the same, the inputs from different training batches are constantly changing. In this scenario, the model is less prone to overfitting the noise from pseudo-labels. Additionally, compared to the teacher model, the student model can obtain more comprehensive representations. Moreover, the randomly augmented images are supervised by the same pseudo-label, which means that consistency regularization is applied to the same unlabeled image across different training batches. Therefore, the self-training approach with strong data augmentation combines two methods: entropy minimization and consistency regularization, commonly used in semi-supervised learning. At this point, the unsupervised objective can be formalized as follows:

$$L^u = \mathrm{H}(T(x), S(A^s(A^w(x)))) \tag{5}$$

where $A^s$ denotes the application of strong data augmentation to the unlabeled data.

The approach employed in this study eliminates the need to select high-quality pseudo-labels based on a threshold. Moreover, it only requires the construction of pseudo-labels once for the unlabeled images, without the need for iterative regeneration of pseudo-labels by the student model. This enables training to proceed in a fully supervised manner, saving training time. The pseudocode for the semi-supervised self-training is presented in Algorithm 1 [31], while the segmentation process is illustrated in Figure 3.

---

**Algorithm 1** Self-training pseudocode

---

**Input:** Labeled training set $D^l = \{(x_i, y_i)\}_{i=1}^{M}$
        Unlabeled training set $D^u = \{u_i\}_{i=1}^{N}$
        Weak/Strong data augmentations $A^w/A^s$
        Teacher/Student model $T/S$
**Output:** Student model $S$
Train $T$ on $D^l$ with cross-entropy loss $L_{ce}$
Obtain pseudo labeled $\hat{D}^u = \{(u_i, T(u_i))\}_{i=1}^{N}$
Over-sample $D^l$ to around the sized of $\hat{D}^u$
**for** *minbatch* $\{(x_k, y_k)\}_{k=1}^{B} \subset (D^l \cup \hat{D}^u)$ **do**
    **for** $k \in \{1, \cdots, B\}$ **do**
        **if** $k \in \{1, \cdots, B\}$ **then**
            $x_k, y_k \leftarrow A^s(A^w(x_k, y_k))$
        **else**
            $x_k, y_k \leftarrow A^w(x_k, y_k)$
        $\hat{y}_k = S(x_k)$
    Update $S$ to minimize $L_{ce}$ of $\{(\hat{y}_k, y_k)\}_{k=1}^{B}$
return $S$

---

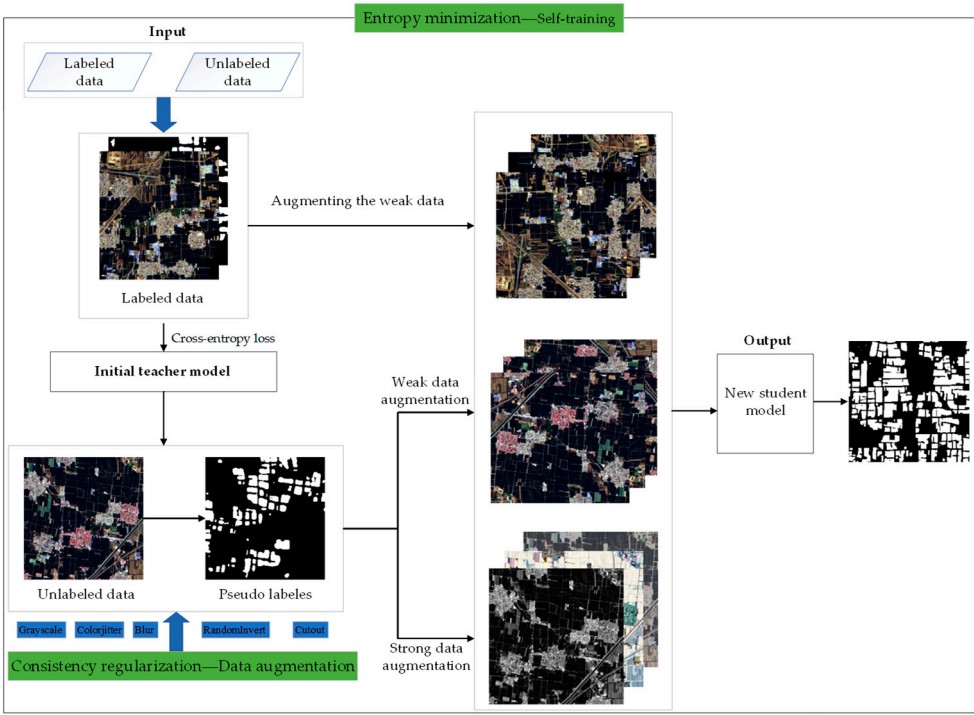

**Figure 3.** Workflow chart of the semi-supervised classification.

## 2.6. Evaluation Metrics and Comparative Methods

Semantic segmentation is considered a multi-classification problem, with mean pixel accuracy (MPA) and mean intersection over union (MIoU) serving as the evaluation metrics for wheat segmentation [32]. A higher value for these metrics indicates better segmentation performance for the model. Let $k$ denote the number of object categories that can be segmented in the dataset, with a total of $k + 1$ categories, where 1 represents the background. This experiment includes two categories: wheat and background. Let $p_{ij}$ denote the number of pixels belonging to category $i$ that are incorrectly classified as category $j$, $p_{ii}$ denote the number of pixels correctly classified as category $i$, and $p_{jj}$ denote the number of pixels correctly classified as category $j$.

Pixel accuracy (PA) is the ratio of correctly classified wheat and background pixels to the total number of pixels in the image. Mean PA (MPA) is an improved version of PA

that calculates the prediction accuracy of wheat and background pixels separately, taking the average of the two to better reflect the overall semantic segmentation accuracy of the model. It is defined as follows:

$$\text{MPA} = \frac{1}{k+1} \times \sum_{i=0}^{k} \frac{p_{ii}}{\sum_{j=0}^{k} p_{ij}} \tag{6}$$

MIoU is a commonly used metric for evaluating the effectiveness of image semantic segmentation. It calculates the ratio between the intersection and union of the true and predicted values, measuring the degree of similarity between the predicted result and the ground truth. A value closer to 1 indicates a more accurate prediction. It is defined as follows:

$$\text{MIoU} = \frac{1}{k+1} \sum_{i=0}^{k} \frac{P_{ii}}{\sum_{j=0}^{k} p_{ij} + \sum_{j=0}^{k} p_{ji} - p_{ii}} \tag{7}$$

To provide a comparative analysis with fully supervised semantic segmentation, three segmentation networks, namely SegNet [33], U-Net [34], and DeepLabv3+ [35], were selected for evaluating the results. SegNet is a deep, fully convolutional neural network architecture for semantic pixel-wise segmentation, which consists of an encoder network and a corresponding decoder network, followed by a pixel-wise classification layer. The fundamental concept of U-Net lies in the integration of skip connections, resulting in a substantial enhancement in image segmentation accuracy, which comprises three primary components: the decoder, encoder, and bottleneck layer. In contrast to SegNet and U-Net, the primary distinguishing feature of DeeplabV3+ lies in its incorporation of dilated convolutions. This augmentation, without compromising information integrity, expands the receptive field, enabling each convolutional output to encompass a broader range of information. This facilitates the extraction of multiscale information, thus enhancing the model's semantic segmentation capabilities. All three models were trained and tested using the same data, and the evaluation metrics employed were MPA and MIoU.

## 3. Results

### 3.1. Comparison of Segmentation Accuracies

The experimental training data consisted of remote sensing images from Zhengding County captured by Landsat-8 OLI. A random sampling strategy was employed to extract 1/16, 1/8, 1/4, and 1/2 proportions of labeled data, resulting in datasets with 312, 625, 1250, and 2500 original images, respectively. The remaining 4688, 4375, 3750, and 2500 images were used as unlabeled data. Initially, the labeled data in different proportions were fed into the segmentation networks to train the teacher models. Subsequently, pseudo-labels were generated for the unlabeled data using the teacher models, and a dataset combining pseudo-labeled and labeled data was created for training the student models. This process was repeated for each of the four datasets with different proportions of labeled data. Finally, the student models were trained using the semi-supervised approach. To reduce the error caused by manual labeling, random sampling of 1/16, 1/8, 1/4, and 1/2 proportions of the labeled data was performed three times on the original 5000 images, and the average results of the three runs were reported. The test set for all experiments was the Zengcun town image in Gaocun, Dingxing County, captured by Landsat-8 OLI. The evaluation metrics for the three segmentation networks in the semi-supervised setting are presented in Table 1.

**Table 1.** Comparison of the semi-supervised classification effects of different methods.

| Method | Indicator | 1/16(312) | 1/8(625) | 1/4(1250) | 1/2(2500) |
|---|---|---|---|---|---|
| Fully supervised SegNet with only the labeled data | MPA/% | 61.82 | 68.87 | 76.95 | 78.17 |
| | MIoU/% | 50.26 | 62.52 | 68.19 | 70.18 |
| Semi-supervised SegNet | MPA/% | 66.01 | 73.77 | 79.66 | 80.51 |
| | MIoU/% | 54.81 | 66.46 | 72.28 | 73.48 |
| Fully supervised DeepLabv3+ with only the labeled data | MPA/% | 78.29 | 79.33 | 81.14 | 82.27 |
| | MIoU/% | 71.82 | 72.48 | 73.39 | 74.49 |
| Semi-supervised DeepLabv3+ | MPA/% | 79.25 | 80.60 | 82.13 | 82.50 |
| | MIoU/% | 73.45 | 74.29 | 75.30 | 75.74 |
| Fully supervised U-Net with only the labeled data | MPA/% | 81.63 | 82.88 | 83.99 | 84.62 |
| | MIoU/% | 73.31 | 74.11 | 75.73 | 76.45 |
| Semi-supervised U-Net | MPA/% | 82.50 | 84.60 | 84.27 | 85.52 |
| | MIoU/% | 76.01 | 76.84 | 77.52 | 77.83 |

*3.2. Visualization Effects Using Different Models*

This paper presents a semi-supervised approach that combines self-training with unlabeled data augmentation to analyze the experimental results of three semantic segmentation networks: SegNet, DeepLabv3+, and U-Net. Initially, only 1/16, 1/8, 1/4, and 1/2 of the labeled data are used to train the SegNet, DeepLabv3+, and U-Net segmentation networks in a fully supervised manner, resulting in the experimental results shown in Figure 4, Figure 6 and Figure 8, and Table 1 (annotated as the fully supervised section). The remaining unlabeled data are then used for self-training in a semi-supervised manner using the SegNet, DeepLabv3+, and U-Net segmentation networks. Two representative result images are selected: one with a higher density of wheat and another with a lower density, along with a higher presence of buildings and bare soil areas. The fully supervised and semi-supervised prediction results of SegNet with different proportions of labeled data are shown in Figures 4 and 5, respectively. The fully supervised and semi-supervised prediction results of DeepLabv3+ with different proportions of labeled data are shown in Figures 6 and 7, respectively. The fully supervised and semi-supervised prediction results of U-Net with different proportions of labeled data are shown in Figures 8 and 9, respectively. From the prediction results, it can be observed that as the proportion of labeled data increases, the model learns better, and the predicted result images show superior performance.

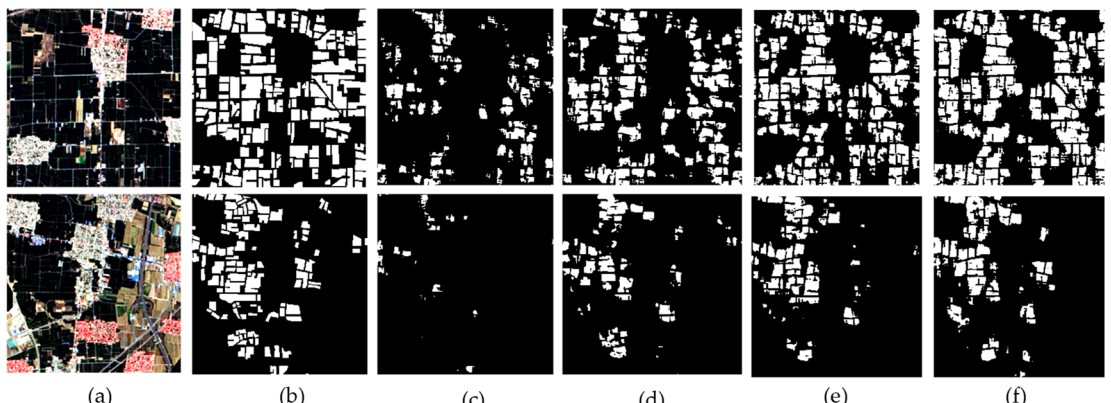

(a)          (b)          (c)          (d)          (e)          (f)

**Figure 4.** Comparison of extraction effects for the fully supervised SegNet under different ratios of labeled data: (**a**) Original image; (**b**) labeled data; (**c**) 1/16; (**d**) 1/8; (**e**) 1/4; and (**f**) 1/2.

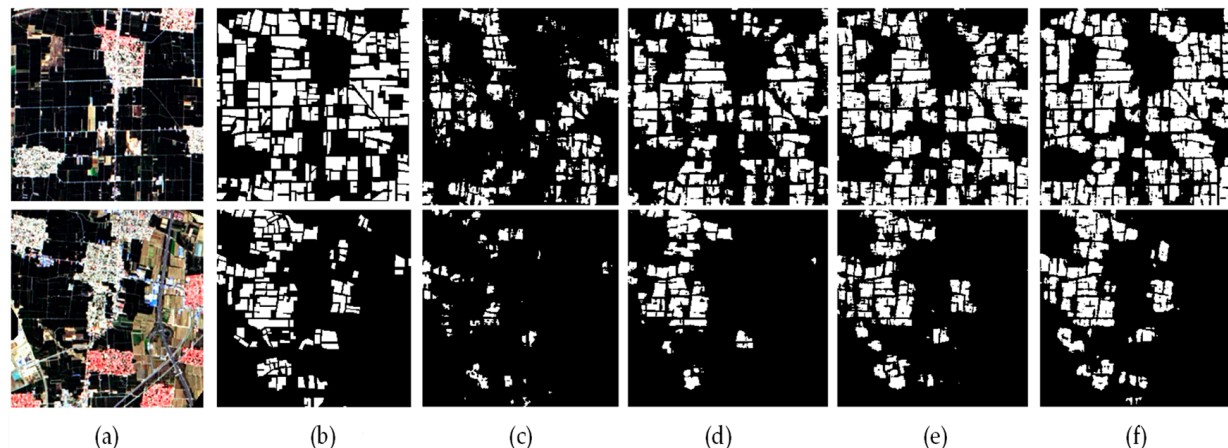

**Figure 5.** Comparison of extraction effects for the semi-supervised SegNet under different ratios of labeled data: (**a**) Original image; (**b**) labeled data; (**c**) 1/16; (**d**) 1/8; (**e**) 1/4; and (**f**) 1/2.

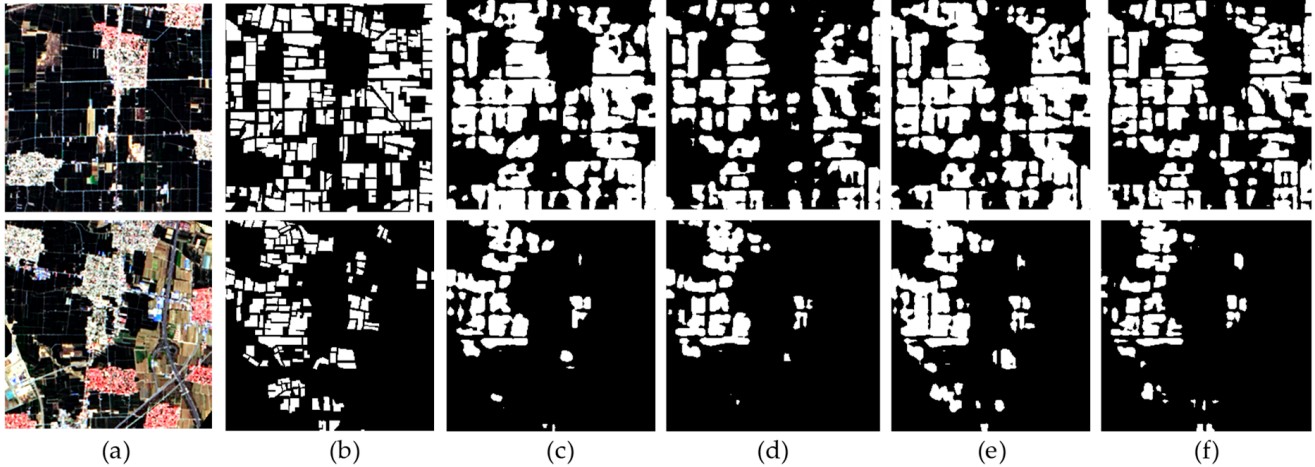

**Figure 6.** Comparison of extraction effects for the fully supervised DeepLabv3+ under different ratios of labeled data: (**a**) Original image; (**b**) labeled data; (**c**) 1/16; (**d**) 1/8; (**e**) 1/4; and (**f**) 1/2.

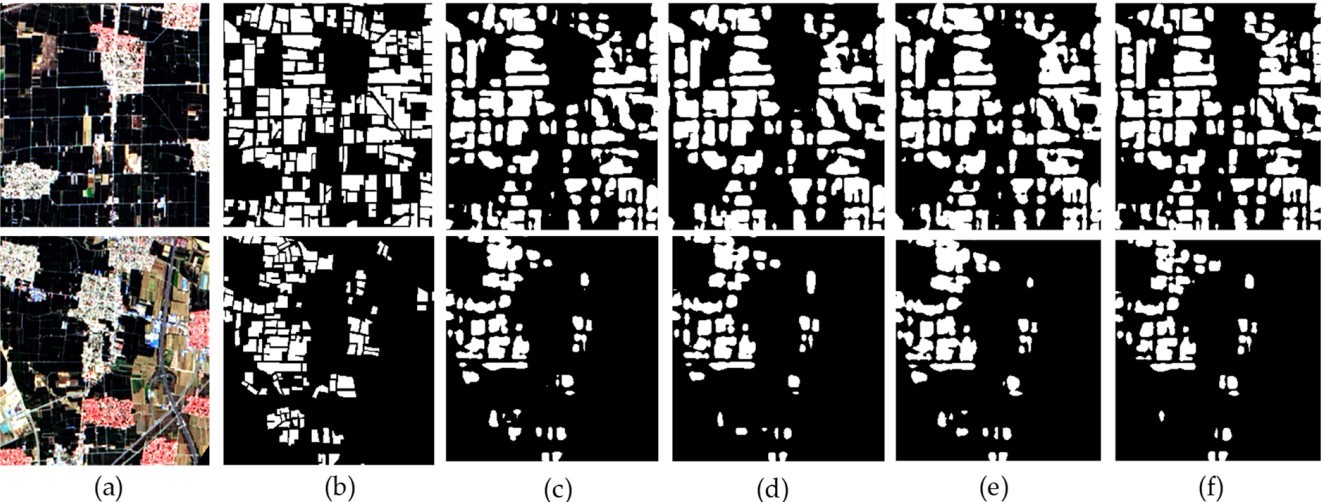

**Figure 7.** Comparison of extraction effects for the semi-supervised DeepLabv3+ under different ratios of labeled data: (**a**) Original image; (**b**) labeled data; (**c**) 1/16; (**d**) 1/8; (**e**) 1/4; and (**f**) 1/2.

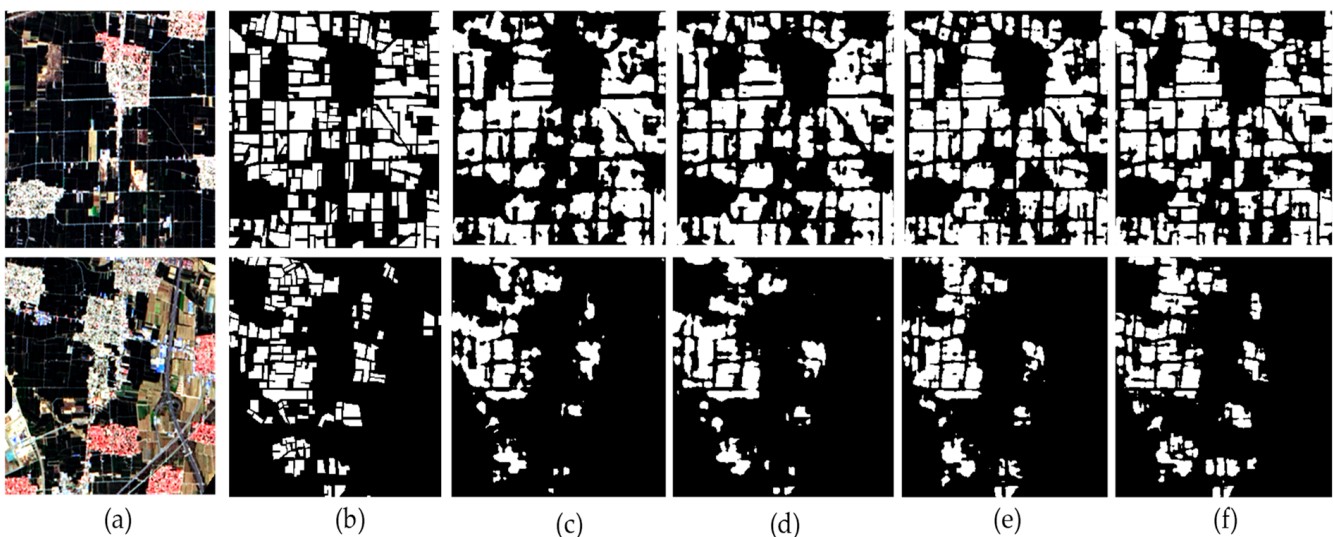

**Figure 8.** Comparison of extraction effects for the fully supervised U-Net under different ratios of labeled data: (**a**) Original image; (**b**) labeled data; (**c**) 1/16; (**d**) 1/8; (**e**) 1/4; and (**f**) 1/2.

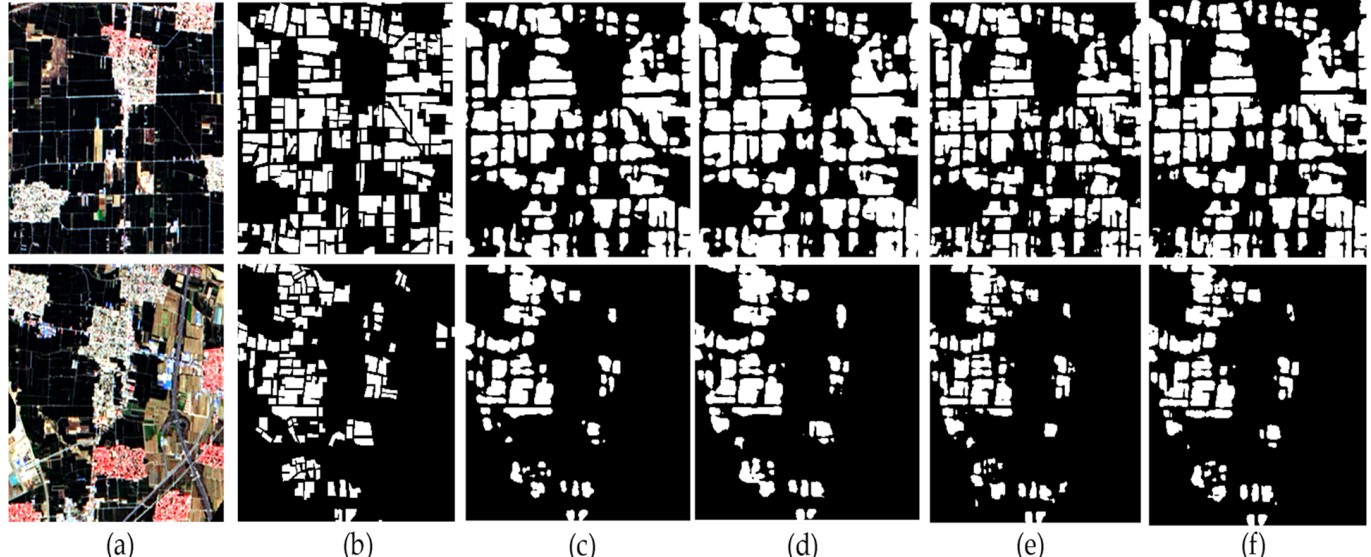

**Figure 9.** Comparison of extraction effects for the semi-supervised U-Net under different ratios of labeled data: (**a**) Original image; (**b**) labeled data; (**c**) 1/16; (**d**) 1/8; (**e**) 1/4; and (**f**) 1/2.

Overall, this study demonstrates the superiority of the semi-supervised self-training method, which incorporates unlabeled data with strong data augmentation, over fully supervised learning methods when training models with different proportions of labeled data. Particularly, the effectiveness of the semi-supervised approach is more pronounced in scenarios with limited labeled data, such as when only 1/16 of the data are labeled. For SegNet, the fully supervised prediction model achieves an MPA of 61.82% and an MIoU of 50.26%, while the semi-supervised prediction model achieves an MPA of 66.01% and an MIoU of 54.81%. For DeepLabv3+, the fully supervised prediction model achieves an MPA of 78.29% and an MIoU of 71.82%, while the semi-supervised prediction model achieves an MPA of 79.25% and an MIoU of 73.45%. For U-Net, the fully supervised prediction model achieves an MPA of 81.63% and an MIoU of 73.31%, while the semi-supervised prediction model achieves an MPA of 82.50% and an MIoU of 76.01%. Comparing the semi-supervised results with the fully supervised results, both the MPA and MIoU show improvement, enabling the attainment of superior solutions even with insufficient labeled data.

### 3.3. Mapping Wheat Planting Areas Using the Semi-Supervised U-Net

Based on the analysis of experimental results, the semi-supervised method employed in this study demonstrates improvements in training with labeled data proportions of 1/16, 1/8, 1/4, and 1/2. Furthermore, this method is applicable to three different semantic segmentation networks: SegNet, DeepLabv3+, and U-Net. Among these networks, U-Net exhibits superior overall prediction performance in semantic segmentation. Figure 10 illustrates the overall results of semi-supervised prediction using different proportions of labeled data for the Landsat-8 OLI image in the rural area. It can be observed from the figure that when only 1/16 of the data are labeled for fully supervised segmentation, a significant portion of the wheat areas remain unidentified. However, with the application of semi-supervised semantic segmentation, there is a noticeable improvement in wheat segmentation.

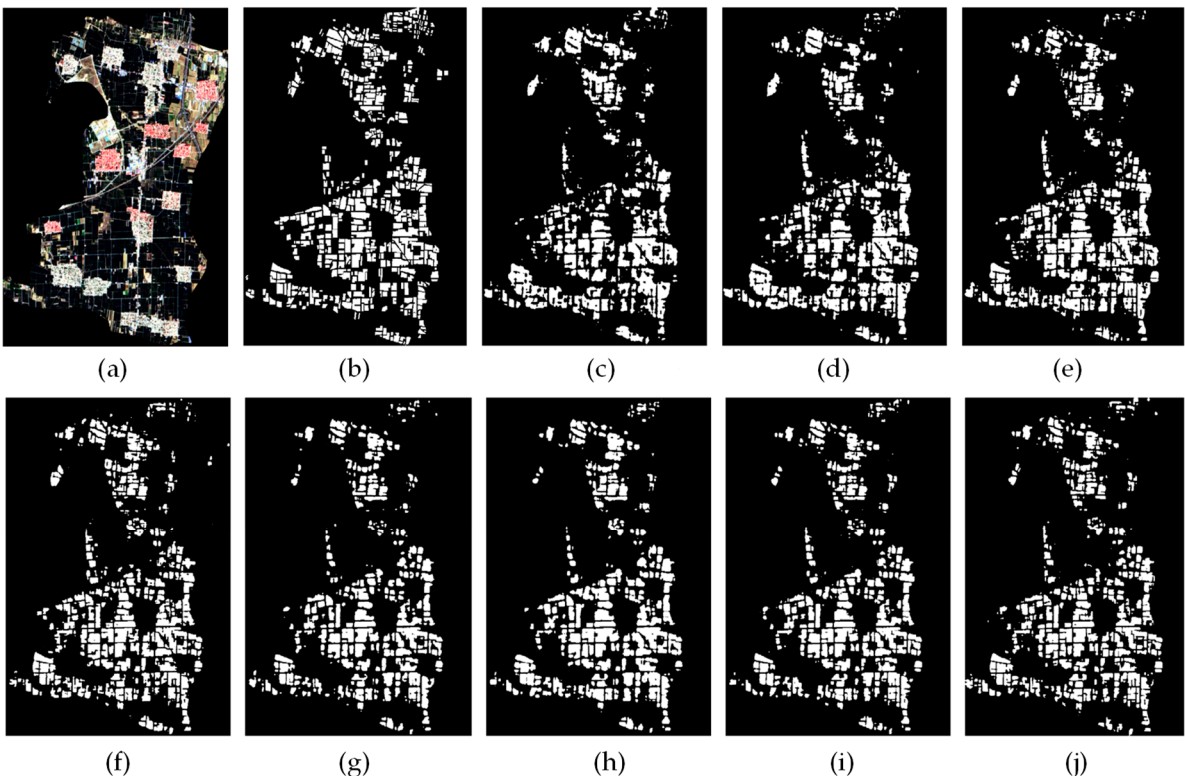

**Figure 10.** Comparison of extraction effects for the semi-supervised U-Net under different ratios of labeled data: (**a**) Original image; (**b**) labeled data; (**c**) 1/16 fully supervised; (**d**) 1/8 fully supervised; (**e**) 1/4 fully supervised; (**f**) 1/2 fully supervised; (**g**) 1/16 semi-supervised; (**h**) 1/8 semi-supervised; (**i**) 1/4 semi-supervised; and (**j**) 1/2 semi-supervised.

## 4. Discussion

### 4.1. Analysis of Training Ratios of Labeled Data for Different Models

In the case of SegNet, when the labeled data are scarce, the segmentation results are poor, with most wheat areas remaining unrecognized, especially in areas with fewer wheat crops. Even with only 1/16 of the labeled data used for training, the generated prediction model is almost unable to identify the wheat areas. However, as the proportion of labeled data increases, the segmentation improves. From the experimental results, it can be seen that the semi-supervised approach improves the segmentation results for different proportions of labeled data, with a significant increase in the predicted area. In the case of DeepLabv3+, when the labeled data are scarce, the wheat areas are often connected in large clusters, making it difficult to distinguish them from other non-wheat areas and resulting in misclassifications. By comparing the fully supervised prediction results of 1/16 and 1/2 labeled data, it can be observed that increasing the amount of labeled data partially alleviates the issue of connected prediction regions. In contrast, the

semi-supervised approach, based on the fusion of labeled and unlabeled data, effectively mitigates the problem of connected prediction regions caused by limited labeled data and performs better in terms of prediction accuracy compared to using only labeled data [36]. U-Net performs better than SegNet and DeepLabv3+ in predicting wheat planting areas. Comparing the results with fully supervised learning, a smaller amount of training data leads to less information learned by the model. The semi-supervised self-training method used in this paper improves performance in various proportions of labeled data compared to fully supervised learning.

### *4.2. Influence of Spatial Resolution on Remote Imagery*

The spatial resolution of remote sensing imagery plays a crucial role in object segmentation and extraction [37]. Higher spatial resolution allows for the identification of smaller objects in an image. Fine details and intricate features of objects can be captured more accurately with higher spatial resolution imagery. This is particularly important when dealing with small and closely spaced objects, such as individual trees in a forest or vehicles in an urban area. Spatial resolution also affects the ability to accurately delineate the boundaries of objects. Higher spatial resolution imagery provides sharper and better-defined edges, enabling more precise and accurate object segmentation. This is particularly important when dealing with objects that have complex or irregular shapes, such as rivers, coastlines, or land-use boundaries.

Nevertheless, it is not necessarily better to have a higher spatial resolution in remote sensing imagery. On the contrary, it is important to consider the distribution range, spatial aggregation characteristics, area coverage, and contrast with the background features of the objects to be segmented and extracted in the image [38]. Sometimes, it is necessary to consider the efficiency of object extraction as well as the cost of acquiring the imagery. Since the selected experimental data are based on Landsat-8 OLI imagery with a resolution of 15 m, the wheat labeling map reveals that manually annotated wheat areas often appear as small, fragmented patches, especially in areas where wheat is interspersed with buildings or bare soil. For instance, in the upper right corner of Figure 10b, the labeling is not well defined, which may lead to biased predictions and instances of missed segmentation. Nevertheless, this approach helps to reduce the cost of acquiring high-resolution remote sensing imagery. Overall, the wheat areas show minimal deviation in terms of their spatial distribution, thereby validating the reliability of the prediction results.

## 5. Conclusions

The study combines entropy minimization and consistency regularization to perform semi-supervised learning. Three mainstream segmentation networks, namely SegNet, DeepLabv3+, and U-Net, are employed to comparatively identify wheat planting areas using semi-supervised semantic segmentation. The labeled data are randomly sampled at proportions of 1/16, 1/8, 1/4, and 1/2, and both fully supervised and self-training semi-supervised methods are employed to train models. A comparative analysis is conducted on the same test set, validating the effectiveness of the semi-supervised approach for different proportions of labeled data and different semantic segmentation networks. The segmentation evaluation metrics, MPA and MIoU, demonstrate varying degrees of improvement, mitigating to some extent the challenge of inadequate labeled data leading to subpar segmentation results. In future research, we will delve into the realm of unsupervised semantic segmentation methods, aiming to automatically extract label data and classification features from the original input imagery for model training purposes.

**Author Contributions:** J.Z. and Y.Z. conceived and designed the experiments; M.Z. and Y.X. performed the experiments; M.Z. and Y.Z. analyzed the data; J.Z. wrote and proofread the paper. All authors have read and agreed to the published version of the manuscript.

**Funding:** This research was funded by the National Natural Science Foundation of China (42301103) and the Industry-University Collaborative Education Project of the Ministry of Education (220802313205840, 202102245009, 22087106262449).

**Data Availability Statement:** Data are contained within the article.

**Acknowledgments:** We gratefully acknowledge the anonymous reviewers for their valuable comments that helped to considerably improve the manuscript.

**Conflicts of Interest:** The authors declare no conflict of interest.

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
