# Peer review of "Semi-Supervised Semantic Segmentation-Based Remote Sensing Identification Method for Winter Wheat Planting Area Extraction"

_agronomy, doi:10.3390/agronomy13122868_

Round 1
Reviewer 1 Report
Comments and Suggestions for Authors

Comments on the Quality of English Language
The quality of the English language is good enough for the manuscript.
Reviewer 2 Report
Comments and Suggestions for Authors
The authors present an interesting paper proposing a Semi-Supervised Semantic Segmentation-Based method for extracting winter wheat planting areas. However, it is advisable to improve some aspects of the paper, namely:
1. - In the literature review, they should give more detail about the methods they use as comparative, as well as identify the limitations/constraints of each of them. Furthermore, substantiates the gains from its application to the case study.
2.- During the description of the method/approach presented by the authors, it would be interesting if they included a "structure of fusion architecture", or "the overall framework model".
3.- Furthermore, it is not clear when it is recommended to use the MPA and MIoU methods/techniques, is it related to some previously defined threshold "between each segment of image"? Is your application dependent on entropy?
Finally, congratulations on your work!
